# How Important Is a Neuron?

**Kedar Dhamdhere**
Google AI
kedar@google.com

**Mukund Sundararajan**
Google AI
mukunds@google.com

**Qiqi Yan**
Google AI
qiqiyan@google.com

## ABSTRACT

The problem of attributing a deep network's prediction to its *input/base* features is well-studied (cf. Simonyan et al. (2013)). We introduce the notion of *conductance* to extend the notion of attribution to understanding the importance of *hidden* units.

Informally, the conductance of a hidden unit of a deep network is the *flow* of attribution via this hidden unit. We can use conductance to understand the importance of a hidden unit to the prediction for a specific input, or over a set of inputs. We justify conductance in multiple ways via a qualitative comparison with other methods, via some axiomatic results, and via an empirical evaluation based on a feature selection task. The empirical evaluations are done using the Inception network over ImageNet data, and a convolutional network over text data. In both cases, we demonstrate the effectiveness of conductance in identifying interesting insights about the internal workings of these networks.

## 1 BACKGROUND AND MOTIVATION

The problem of attributing a deep network's prediction to its input is well-studied (cf. Baehrens et al. (2010); Simonyan et al. (2013); Shrikumar et al. (2017); Binder et al. (2016); Springenberg et al. (2014); Lundberg & Lee (2017); Sundararajan et al. (2017)). Given a function $F : \mathsf{R}^n \to [0, 1]$ that represents a deep network, and an input $x = (x_1, \ldots, x_n) \in \mathsf{R}^n$, an attribution of the prediction at input $x$ is $(a_1, \ldots, a_n) \in \mathsf{R}^n$ where $a_i$ can be interpreted as the *contribution* of $x_i$ to the prediction $F(x)$. For instance, in an object recognition network, an attribution method could tell us which pixels of the image were responsible for a prediction. Attributions help determine the influence of base features on the input.

Many papers on deep learning speculate about the importance of a hidden unit towards a prediction. They tend to use the activation value of the hidden unit or its product with the gradient as a proxy for feature importance. As we discuss later in the paper, both measures have undesirable behavior. For instance, a ReLU always has positive activation values but could either have positive or negative influence on the prediction; the fact that the sign of the influence cannot be identified is undesirable. In the next subsection, we introduce our approach to the problem of neuron importance. We compare it to the related work in Secion 2.

### 1.1 OUR CONTRIBUTION

We introduce the notion of conductance (see Section 3) based on a previously studied attribution technique called Integrated Gradients (Sundararajan et al. (2017)). Integrated Gradients rely on integrating (summing) the gradients (of the output prediction with respect to the input) over a series of carefully chosen variants of the input. Informally, the conductance of a hidden unit of a deep network is the *flow* of Integrated Gradients' attribution via this hidden unit. The key idea behind conductance is to decompose the computation of Integrated Gradients via the *chain rule* (see Equations 2 and 3 for formal definitions).

We qualitatively compare conductance to other commonly used measures of neuron importance in Section 4.1. Conductance takes into account the activation of the neuron, the partial derivative of a the output with respect to the neuron and the partial derivative of the neuron with respect to the input. Intuitively, all three quantities play an important role in judging the importance of a neuron. We find that other measures of neuron importance, like activation values and gradients, ignore one or more of these terms and this results in intuitively undesirable attributions.

In Section 4.2, we provide theoretical justification for conductance. To do this, we first state some desirable properties (axioms) of neuron importance. We show that neuron importances that satisfy these very natural axioms *must* be derived from a "path method", i.e., as a path integral of the gradients from the output to the hidden layer (Theorem 1). Furthermore, under an additional condition of consistency to the base feature attributions, we are able to show that the path taken in the hidden layer must depend on the layers below (Theorem 2). Conductance naturally couples the path at the base features with the path at the hidden layer and we observe that it satisfies all of these axioms (Theorem 3).

We also evaluate the effectiveness of conductance (against several related methods) empirically. We do a feature selection study to see if the high-conductance units are also highly predictive. The premise is that if some neurons are consistently important across instances of a certain class, then using these neurons as features, we should be able to classify other instances of the same class. Our empirical evaluations are done over the following two large-scale networks, the Inception architecture (Szegedy et al. (2014)) trained on ImageNet data (Russakovsky et al. (2015)) and a highly-cited convolution network (Kim (2014a)) frequently used for text classification tasks.

Though all our claims are about importance, and not about interpretability, we acknowledge that the main application of studying neuron importance is to interpret the function of the neuron. In this regard, we demonstrate that conductance can give us intuitive insights about the network. For instance, for the object detection task, we identify filters that are influential *across* images and labels, which seem to correspond to texture or color features. And for a sentiment analysis task, we use conductance to identify neurons that detect negation.

## 2 RELATED WORK

A recent paper (Datta et al. (2018)) is closely related to ours in intention and technique. It proposes a notion of influence of an hidden neuron on the output prediction. We explicitly compare our technique to theirs, both qualitatively and quantitatively. We show for instance, that their technique can sometimes result in importance scores with incorrect signs.

There is also a large literature on understanding the function/operation of hidden units of a deep network (cf. Mahendran & Vedaldi (2015); Dosovitskiy & Brox (2015); Yosinski et al. (2015); Mordvintse et al. (2015; 2017)). These approaches are usually optimization based, i.e. they explicitly tweak images to optimize activations of a neuron or a group of neurons. The resulting images are an indicator of the function of the hidden unit. However, these works don't identity the importance of a hidden unit (either to a prediction or to a class of predictions). Also, these approaches tend not to work for every hidden unit. Consequently, it is not clear if all the hidden units which are important to the predictive task have been investigated by these approaches. Another approach to identifying the function of hidden units is to build a simple, linear, explanatory model (cf. Le et al. (2012); Alain & Bengio (2016)) in parallel to the original model. But then it is no longer clear if the influence of the hidden units in the new model is the same as what it is in the original model; e.g. if there are correlated hidden neurons, their weights in the explanatory model may not match those in the original model. Zhou et al. (2018) approach the interpretability of hidden neurons by aligning human-labeled concepts with the hidden neurons by using neuron activations for each concept. Finally, Kim et al. (2018) takes a human-defined concept as input, finds hidden neurons that are predictive of this concept, and identifies the influence of these neurons on a certain prediction. Because the network may rely on an inherently different representation from the human, it is not clear how this process identifies all the concepts important to the prediction for a given input or a class of inputs. In contrast, we rely on the network's own abstraction, i.e., the filters in the hidden layers. This will result in accounting for the whole prediction, but may possibly result in less intuitive explanations.

## 3 CONDUCTANCE

In this section, we define conductance formally. We start with some notation. Suppose we have a function $F : \mathsf{R}^n \to [0, 1]$ that represents a deep network. Specifically, let $x \in \mathsf{R}^n$ be the input at hand. We will define attribution methods *relative* to a baseline input $x' \in \mathsf{R}^n$.[1] For image networks, the baseline could be the black image, while for text models it could be the zero embedding vector.

We now recall the definition of Integrated Gradients. We consider the straight-line path (in $\mathsf{R}^n$) from the baseline $x'$ to the input $x$, and compute the gradients at all points along the path. Integrated gradients are obtained by cumulating these gradients. Specifically, integrated gradients are defined as the path integral of the gradients along the straight-line path from the baseline $x'$ to the input $x$. Here is the intuition for why the method works for, say, an object recognition network. The function $F$ varies from a near zero value for the informationless baseline to its final value. The gradients of $F$ with respect to the image pixels explain each step of the variation in the value of $F$. The integration (sum) over the gradients cumulates these micro explanations and accounts for the net difference between the baseline prediction score (near zero) and the prediction value at the input $x$.

Formally, the integrated gradient for the $i^{th}$ base feature (e.g. a pixel) of an input $x$ and baseline $x'$ is:

$$\mathsf{IG}_i(x) ::= (x_i - x_i') \cdot \int_{\alpha=0}^{1} \frac{\partial F(x' + \alpha(x - x'))}{\partial x_i} \, d\alpha \qquad (1)$$

where $\frac{\partial F(x)}{\partial x_i}$ is the gradient of $F$ along the $i^{th}$ dimension at $x$.

Notice that Integrated Gradients produces attributions for base features (e.g. the pixels of an object recognition network). There is a natural way to 'lift' these attributions to a neuron in a hidden layer. Consider a specific neuron $y$ in a hidden layer of a network. We can define the conductance of neuron $y$ for the attribution to an input variable $i$ as:

$$\mathsf{Cond}_i^y(x) ::= (x_i - x_i') \cdot \int_{\alpha=0}^{1} \frac{\partial F(x' + \alpha(x - x'))}{\partial y} \cdot \frac{\partial y}{\partial x_i} \, d\alpha \qquad (2)$$

We can define the total conductance of the hidden neuron $y$ by summing over the input variables:

$$\mathsf{Cond}^y(x) ::= \sum_i (x_i - x_i') \cdot \int_{\alpha=0}^{1} \frac{\partial F(x' + \alpha(x - x'))}{\partial y} \cdot \frac{\partial y}{\partial x_i} \, d\alpha \qquad (3)$$

We will interchangeably use the term "conductance of neuron $j$" to mean either of Equations 2 or 3. We use the former to explain the function of the neuron in terms of its effect on the base features of the input. We use the latter to discuss the importance of the neuron.

Frequently, we will aggregate over a set of logically related neurons that belong to a specific hidden layer. For instance, these could be neurons that belong to a single filter. In this case, we will sum over the conductances of the neurons in the set to define the conductance of the set as whole.

**Remark 1.** *There is a different way to extend Integrated Gradients for neuron importance. The idea is to treat the neuron activations in the hidden layer under consideration as inputs of the network from that layer onwards. Then integrated gradients can be applied at this layer. This idea is described in (Shrikumar et al. (2018)).*

*A significant disadvantage of this approach is that it has no obvious analog to Equation 2, which provides an interpretation of the neuron importance at the level of the base features, which we need for interpretability. However, a possible advantage is in computation: If we were only interested in Equation 3, then this approach can be computed more efficiently than conductance in standard deep-learning platforms like tensorflow or pytorch, because it can be computed by a single back-prop (per interpolation step).*

---

[1]Several attribution methods (cf. Shrikumar et al. (2017); Binder et al. (2016); Sundararajan et al. (2017)) produce attributions *relative* to an extra input called the baseline. All attribution methods rely on some form of sensitivity analysis, i.e., perturbing the input or the network, and examining how the prediction changes with such perturbations. The baseline helps defines these perturbations.

# 4 EVALUATION OF CONDUCTANCE

We compare conductance against three other methods. The first two are commonly used in literature and the third is from a recent paper (Datta et al. (2018)).

- **Activation:** The value of the hidden unit is the feature importance score.
- **Gradient×Activation:** $y \times \frac{\partial F(x' + \alpha \times (x - x'))}{\partial y}$
- **Internal Influence:** (Datta et al. (2018)) The measure of feature importance is:

$$\mathsf{IntInf}^y(x) ::= \int_{\alpha=0}^{1} \frac{\partial F(x' + \alpha \times (x - x'))}{\partial y} \, d\alpha \tag{4}$$

It is notoriously hard to compare feature importance methods. We provide three types of justifications for our method to address this. First, we qualitative issues with other methods (Section 4.1). These issues don't affect Conductance. Second, we provide an axiomatic justification of Conductance (Section 4.2). Third, we use the importance scores computed via conductance and the other methods within a feature selection task. The premise is that hidden units that are important across a set of inputs from a class should be predictive of this input class (Sections 5 and 6).

## 4.1 PROBLEMS WITH OTHER METHODS

Saturation of neural networks is a central issue that hinders the understanding of nonlinear networks. This issue was discussed in detail by Sundararajan et al. (2017) for Integrated Gradients. Basically, for a network, or a sub-network, even when the output crucially depends on some input, the gradient of the output w.r.t. the input can be near-zero.

As an illustrative example, suppose the network first transforms the input $x$ linearly to $y = 2x$, and then transforms it to $z = \max(y, 1)$. Suppose the input is $x = 1$ (where $z$ is saturated at value 1), with 0 being the baseline. Then for the hidden unit of $y$, gradient of $z$ w.r.t. $y$ is 0. Gradient×activation would be 0 for $y$, which does not reflect the intuitive importance of $y$. Like in Integrated Gradients, in computing conductance, we consider all extrapolated inputs for $x$ between 0 and 1, and look at the gradients of output w.r.t. $y$ at these points. This takes the non-saturated region into account, and ends up attributing 1 to $y$, as desired.

We now show that the methods we compare against can yield scores that have signs and magnitudes that are intuitively incorrect (in particular, Axiom 3 from Section 4.2 is violated). This is intuitively because each misses terms/paths that our method considers. Activation values for a ReLU based network are always positive. However, ReLU nodes can have positive or negative influence on the output depending on the upstream weights. Here, Activation does not distinguish the sign of the influence, whereas conductance can.

Gradient×Activation as a linear projection can overshoot. Certain hidden units that actually have near zero influence can be assigned high importance scores. For example, suppose that the network is the composition of two functions $f(x) = x$ and a weighted ReLU $g(y) = max(y - 1, 0)$. Again, the network computes the composition $g(f(x))$. Suppose that the baseline is $x = 0$ and the input is $x = 1 - \epsilon$. The output of the network is $g(f(1 - \epsilon)) = 0$. But the feature importance of the unit $f$ is $1 - \epsilon$ (activation is $1 - \epsilon$ and gradient is 1). Notice that this is the only unit in its layer, so the fact that its influence does not agree in magnitude with the output is undesirable. In contrast, conductance assigns all hidden units a score of zero.

The internal influence (Datta et al. (2018)) can disagree in sign with actual direction of the function. Suppose that the network is the composition two functions $f(x) = -x$ and $g(y) = y$, i.e., the network computes the composition $g(f(x))$. Suppose that the baseline is $x = 0$ and the input is $x = 1$. The output of the network is $-1$. But the internal influence of the unit represented by the function $g$ is $+1$ (regardless of the choice of the input or the path). Notice that this is the only unit in its layer, so the fact that its influence does not agree in sign with the output is highly undesirable. In contrast, conductance assigns an influence score of $-1$.

Finally, neuron importance can also be defined for other attribution methods like DeepLift (Shrikumar et al. (2017)) and Deep SHAP (Lundberg & Lee (2017)). These methods rely on back propagating

activation differences (between the input and the baseline) from the output node towards the input. In this process, they do compute neuron importance for each hidden layer. The critique from Sundararajan et al. (2017) that these methods do not satisfy implementation invariance still applies. See Appendix for an example.

## 4.2 A Partial Axiomatization of Conductance

We have a function $F(x) = g(h(x))$, where $h(x) = y$ represents the hidden layer. We assume that the functions are both continuous and differentiable. Recall that we have an input $x$ and a baseline $x'$ and the corresponding values of the hidden layer $y$ and $y'$.

The conductance expression involves two gradients, the set of partials of the output with respect to the hidden layer $\frac{\partial g}{\partial y}$, and the set of partials of the hidden layer with respect to the input $\frac{\partial h}{\partial x}$. Theorem 1 establishes the unique form of the dependence on the first of the two partials. Theorem 2 shows that a naive approach that treats that hidden layer as the input, for instance applying Integrated Gradients directly at the hidden layer, causes the hidden layer and the base features to have inconsistent attributions.

Define a hidden-layer path method to be an attribution method that is represented in terms of path integrals $\int \frac{\partial g}{\partial y} d\gamma$. Here $\gamma$ is a path in the hidden layer space; i.e. $\gamma$ is a continuous function over $0, \infty]$ such that $\gamma(0) = y'$ and there is a $t'$ such that $\gamma(t) = y$ for $t \geq t'$. Further, the path $\gamma$ can be a function of the input, the baseline and the function $h$, but not of the function $g$.

The result relies on three axioms:

- **1. Linearity for Hidden Features**: Suppose the function $g = a \times g_1 + b \times g_2$, for some scalars $a$ and $b$, then the attributions for the function $g$ are a linear combination of the attributions of $g_1$ and $g_2$ with weights $a$ and $b$ respectively. [2]

- **2. Insensitivity for Hidden Features**: If a certain hidden variable $y_i$ has a uniformly zero partial $\frac{\partial g(y)}{\partial y_i}$ at all values of $y$, then it always gets a zero attributions.

- **3. Completeness for Hidden Features**: The attributions for the hidden layer add up to the difference of the function values at the input and the baseline $F(x) - F(x')$. [3].

With the description of the desired characteristics (axioms) for attribution methods, we state our main theorem:

**Theorem 1.** *For bounded discrete domains, the only attributions methods that satisfy Axioms 1-3 are convex combinations of hidden-layer path methods.*

Here we only prove the statement for cases where the domains involved are discrete and bounded. Gradients for such domains correspond to discrete gradients. A more general proof for the continuous case (the discrete case in the limit as we refine the discretization) seems to require heavy mathematical machinery from functional analysis that is out of the scope of this paper, and of little practical relevance.

The proof is in the Appendix. Our proof modifies the proofs of Wang (1999) and Friedman (2004) in a cost-sharing model. Cost-sharing is restriction of the attribution problem with the baseline as the zero vector, the functions are non-negative and monotone, and the cost-shares are always non-negative. One consequence of these assumptions is that the paths followed are monotone. We remove these restrictions and allow the paths to even be non-monotone; this will be crucial when we make the paths depend on the network below.

Here is some rough intuition for the roles played by the various axioms: First, we can define a basis space for the discrete functions. This basis space consists of functions that are 1 for all inputs larger

---

[2]Linearity is intuitively desirable because if the function is linear in the hidden features, then the attributions also reflect this.

[3]It is easy to show that Conductance satisfies this axiom. An immediate consequence is that conductances also satisfy the *Layerwise Conservation Principle*( Bach et al. (2015)), which says that "a network's output activity is fully redistributed through the layers of a DNN onto the input variables, i.e., neither positive nor negative evidence is lost."( Samek et al. (2015)). None of the three methods we compare against satisfy completeness or layerwise conservation. The bad examples from Section 4.1 show this.

than some fixed input and $0$ for all other inputs. Linearity and Insensitivity together imply that the attributions must be the convex combination of the attributions for the basis functions; the convex combination can include negative weights. The completeness condition establish flow conditions on the differentials, and standard theorems about network flows from integer programming show that the flows decompose into paths.

We now attempt to establish the shape of the path in the hidden layer. Recall that Integrated Gradients method is a path method that uses the path function $\gamma^{IG}(t) = x' + (x - x') \times t$ for $t \in [0, 1]$ and $\gamma^{IG}(t) = x$ for $t \geq 1$. [4] We could have applied integrated gradients treating the hidden layer as an input, or More generally, we could have applied some attribution method treating the hidden layer as an input. But, this would make the attribution at the hidden layer inconsistent (formalized by the partition consistency axiom below) with the ones at the base layer. We show this more generally for all *oblivious path methods*, i.e., methods that employ the same path irrespective of the function $h$. (This is equivalent to saying that we treat the hidden layer as an input because we ignore what is below it.)

**4. Partition Consistency**: If there is a partition $P = P_1, P_2, \ldots, P_k$ of the base features and a partition $Q = Q_1, Q_2, \ldots, Q_k$ of the hidden layer such that for every $j \in 1 \ldots k$, every variable in partition $Q_j$ is *only* a function of the variables in the partition $P_j$, then the sum of the attributions of variables in $Q_j$ is equal to those of $P_j$.

Partition consistency intuitively requires that the attributions are conserved across layers along a partition structure, if one exists.

**Theorem 2.** *No oblivious hidden-layer single path method satisfies partition consistency when the base layer attributions are computed by integrated gradients.*

*Proof.* We will do a proof by contradiction. Fix an oblivious hidden-layer path method $\gamma'$ that satisfies partition consistency. By Theorem 1, the path method does not depend on the function $g$. By obliviousness, it also does not depend on the function $h$ or the input $x$ and the baseline $x'$. So the *s*ame path method must satisfy partition consistency for all choices of inputs, baselines, $f$ and $g$. Let us first fix $h$ to be the identity function, i.e, every variable $y_i$ at the hidden layer is a copy of the variable $x_i$. Each variable is its own partition i.e., $P_i = x_i$ and $Q_i = y_i$. By partition consistency, the attributions for $y_i$ for the path method $\gamma'$ should equal that of $x_i$ for IG. Recall that Theorem 1 from Sundararajan et al. (2017) that IG is the unique single path method (at the base layer) that satisfies certain axioms. In particular, this implies that other single-path methods differ in attributions from the IG attributions for *some* choice of function $g$ and inputs and baselines. So the only way for $\gamma'$ to be partition consistent with IG for all functions $g$ and inputs and baselines is for $\gamma'$ to be equal to $\gamma^{IG}$.

To complete the contradiction, consider functions $h$ and $g$ that are insensitive to all but two of the inputs $x_1, x_2$ at the base layer, and two hidden layer nodes $y_1, y_2$. Further, suppose that $y_1 = h_1(x) = x_1^p$ and $y_2 = h_2(x) = x_2^q$ and $g(y) = y_1^{1/p} \cdot y_2^{1/q}$, for some parameters $p > 0$ and $q > 0$. By construction $P_1 = x_1, Q_1 = y_1, P_2 = x_2, Q_2 = y_2$. Note that $g(h(x)) = x_1 \cdot x_2$ is the same function for all $p > 0, q > 0$. Let's consider a baseline $x' = (0, 0)$ and an input $x = (1, 1)$. The IG attribution for the variable $x_1$ is $1/2$ for all $p > 0, q > 0$. So by partition consistency, the attribution for variable $y_1$ via the hidden layer path $\gamma' = \gamma^{IG}$ should be equal to $1/2$ for all $p > 0, q > 0$.

To complete the contradiction: Fix $p = 2$ and $q = 1$, the hidden-layer path $\gamma^{IG}$ yields an attribution for $y_1$ of $\int y_2 \cdot y_1^{-1/2} dt = \int_0^1 t \cdot t^{-1/2} dt$, which is $2/3$. $\qquad\square$

The point of Theorem 2 is that the path at the hidden-layer should depend on the function $h$ (the network below the hidden layer). The most obvious approach is to *couple* the hidden layer path with the IG path via the function $h$. That is, the hidden layer path is $h(\gamma^{IG}) = h(x' + (x - x') \times t)$ for $t \in [0, 1]$. We do not show that this is the unique approach, but this is arguably the most natural coupling. Integrating along this hidden-layer path yields precisely the conductance formula; compare against Equation 3) and note that $dh(\gamma^{IG})/dt = \sum_i (x_i - x_i') \cdot \frac{\partial y}{\partial x_i}$.

---

[4]It is axiomatized using additional axioms for instance Symmetry. But it is pretty clear that symmetry should also depend on the network below the hidden layer, i.e., the function $h$. The proof for Theorem 2 essentially shows this.

Finally, we can show that conductance satisfies all the desirable properties for neuron importances.

**Theorem 3.** *Conductance satisfies Axioms 1-4.*

We omit the easy proofs.

**Remark 2.** *Theorem 1 uses Axiom 1-3 to prove that any attribution method for neuron importance must be a path method. But it doesn't provide any way of ruling out oblivious paths. The only other axiom (Partition consistency) is limited to specific types of networks. So, another axiom is needed to prove uniqueness.*

## 5 EVALUATION AND INSIGHTS FROM AN OBJECT RECOGNITION MODEL

In this section, we describe both qualitative and empirical evaluation of conductance on an object recognition model. We describe some insights we obtained from analyzing the network using conductance (Figures 1-4). We also compare conductance to other neuron importance methods on a feature selection task (Section 5.2).

The network is built using the GoogleNet architecture Szegedy et al. (2014) and trained over the ImageNet object recognition dataset Russakovsky et al. (2015). For a detailed description of the architecture, we refer the reader to Szegedy et al. (2014). We consider the following hidden layers in the network: `mixed3a`, `mixed3b`, `mixed4a`, `mixed4b`, `mixed4c`, `mixed4d`, `mixed4e`, `mixed5a` and `mixed5b`. These layers are likely to encode composite features of the input pixels, such as edges or shapes.

We consider the filters as a unit of analysis for the purpose of studying conductance. The network we analyze was trained using ImageNet dataset. This dataset has 1000 labels with about 1000 training and 50 validation images per label.

### 5.1 INSIGHTS FROM OBJECT RECOGNITION MODEL

We use conductance as a measure to identify influential filters in hidden layers in the Inception network. Given an input image, we identify the top predicted label. For the *pre-softmax* score for this label, we compute the conductance for each of the filters in each of the hidden layers using Equation 3. For the visualization of the conductance at pixel level, we use Equation 2. The visualization is done by aggregating the conductance along the color channel and scaling the pixels in the actual image by the conductance values. See Figures 4 and 3 for examples of images and filters that have high conductances for the image.

Despite the fact that relatively few filters determine the prediction of the network for each input, we found that some of the filters are shared by more than one images with different labels. Two examples of such filters are shown in Figure 1 and 2.

### 5.2 EVALUATION VIA FEATURE SELECTION

We compare conductance with other methods of neuron importance on a feature selection task. A good method of assigning neuron importance should be able to identify the neurons important to not just an input instance, but to an input class. That is to make statements of the form: "yellowness is a salient feature of all bananas", or "wheels are features of all cars". This eval is somewhat similar to the one in Le et al. (2012), which also builds classifiers from high-level concepts. They do it using an autoencoder with sparsity. Whereas we treats filters in hidden layers as representing high-level features.

In our evaluation task, we chose four sets of five labels each from ImageNet. The first two sets were thematically similar (5 species of dog and 5 types of water vessels) and the other two sets had labels chosen randomly. We picked about 50 images per label from the validation set – we split this into 30 images for training and 20 for eval. For each method of feature importance, we compute an importance value for each filter using the method and aggregated those over training set per label. We pick $k$ filters with highest aggregate value for any label. We then use these $k$ filters to classify images from eval set, by training a simple linear classifier. A method that produces the right notion of feature importance should result in a better predictor because it identifies a better list of features. We have

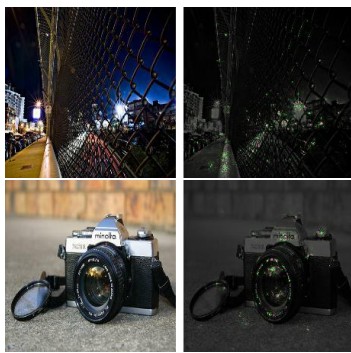

Figure 1: Filter 52 in `mixed3a` layer highlights glare across classes.

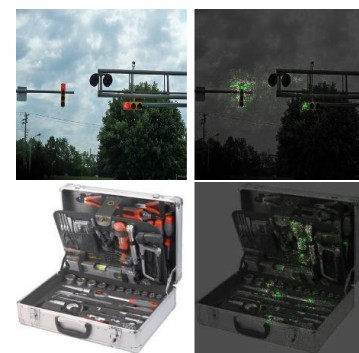

Figure 2: Filter 76 in `mixed4c` layer highlights red colored regions across classes.

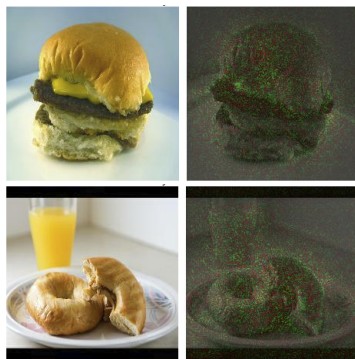

Figure 3: High conductance filter for cheeseburger and bagel classes.

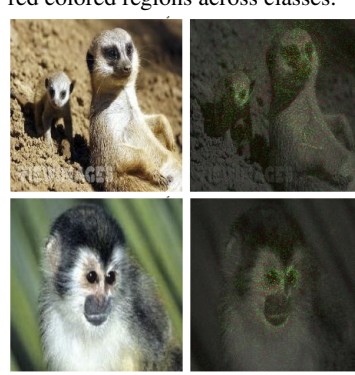

Figure 4: High conductance filter for meerkat and titi classes.

| Method | 5 features | 10 features | 15 features | 20 features | |
|---|---|---|---|---|---|
| activations | 37.80 | 36.59 | 47.56 | 54.88 | |
| gradient*activation | 32.93 | 58.54 | 64.63 | 69.51 | Water vessels task |
| influence | 45.12 | 53.66 | **71.95** | 68.29 | |
| conductance | **48.78** | **68.29** | 70.73 | **79.27** | |
| Method | 5 features | 10 features | 15 features | 20 features | |
| activations | 54.76 | 61.90 | 65.48 | 69.05 | |
| gradient*activation | 67.86 | 83.33 | 85.71 | 88.10 | Random labels task. |
| influence | 50.00 | 67.86 | 73.81 | 75.00 | |
| conductance | **88.10** | **94.05** | **96.43** | **94.05** | |
| Method | 5 features | 10 features | 15 features | 20 features | |
| activations | 46.52 | 48.51 | 55.81 | 60.45 | |
| gradient*activation | 56.94 | 76.01 | 79.74 | 84.39 | Aggregate over 4 tasks. |
| influence | 48.50 | 62.23 | 71.37 | 72.50 | |
| conductance | **68.85** | **81.58** | **85.99** | **87.79** | |

Table 1: Accuracy of classifiers trained on small number of features selected using the four different methods. The first table reports accuracy for classifying between water vessals. The second table shows numbers for classifying between 5 randomly chosen labels (matchstick, meerkat, ruffed grouse, cheeseburger, toaster). The last table reports aggregate over all four label sets.

displayed the results for four methods for these four sets of labels in Table 1. We report results for two label sets: 5 types of water vessels as well as one random label set, as well as aggregate numbers over the four different sets of labels. We observe that conductance leads to better feature selection from these results.

In Figure 4 and 3, we present exemplar images *outside* the training set that have high conductance for two filters that were identified as important for the labels meerkat and cheeseburger respectively.

The first filter appears to be focused on the eyes of Titi and the Meerkat, and the second filter on the break-like surface of the cheeseburger and the bagel.

# 6 EVALUATION AND INSIGHTS FROM A SENTIMENT MODEL

In this section, we do qualitative and empirical evaluations of conductance on text models that use a CNN-based architecture.

First, we analyze a model that scores paragraphs or sentences for sentiment. The model is a convolutional model from Kim (2014b) trained over review data. In the variant of the model we study, the words are embedded into a $50$ dimensional space. The embeddings are fed to a convolutional layer that has $4$ window sizes ranging from $3$ to $6$ words. Each of the four filter widths has $64$ feature maps. Each of the $4 \times 64 = 256$ filters is 1-max pooled to construct a layer of size $256$, which is then fed (fully connected) to a layer of $10$ neurons, which is fed to a logit, that feeds into a softmax that classifies whether the sentiment is positive and/or negative (two binary heads).

We study the conductances of the filters in the convolutional layer. In Section 6.1, we illustrate how conductance can be used to analyze a model and gain insights. As an example, we demonstrate at how positive and negative sentiments are captured in the sentiment model. In Section 6.2, we compare conductance to other neuron importance methods by doing feature selection for a text classification task.

## 6.1 INSIGHTS FROM SENTIMENT MODEL

Negation is commonly used in expressing sentiments, in phrases like "this is not good" or "this is not bad". We apply conductance analysis to answer questions like: does the sentiment network understand negation? Does it have hidden units dedicated to implement the logic of negation? We first identify high conductance filters for the input "this is not good" that have a high attribution to the pattern "not good". We find that three filters fit this pattern. It is possible that these filters perform different functions over other inputs. To show that this is not the case, we then run the network over around $4000$ inputs from the Stanford Sentiment Treebank, noting examples that have a high conductance for any of these three filters. Figure 5 displays the results. Figure 6 has analogous results for the pattern "not bad". However, a different set of filters have a high attribution for "not bad". In the second case, the filters seem quite focused on negation. This suggests that the network does understand negations, and has dedicated filters for handling negations.

The results , if not memorable , are at least interesting .
I ' m sure mainstream audiences will be baffled , but , for those with at least a minimal appreciation of Woolf and Clarissa Dalloway , The Hours represents two of those well spent .
Most consumers of lo mein and General Tso ' s chicken barely give a thought to the folks who prepare and deliver it , so , hopefully , this film will attach a human face to all those little steaming cartons .
An engaging criminal romp that will have viewers guessing just who ' s being conned right up to the finale .
Boasts enough funny dialogue and sharp characterizations to be mildly amusing .
Qutting may be a flawed film , but it is nothing if not sincere .
Conceptually brilliant ... Plays like a living - room War Of The Worlds , gaining most of its unsettling force from the suggested and the unknown .
Made - Up lampoons the moviemaking process itself , while shining a not particularly flattering spotlight on America ' s skin - deep notions of pulchritude .

Figure 5: Sentences with high conductance for filters that have high conductance for the phrase "not good". These filters capture negation, diminishing, and show some stray errors.

## 6.2 EVALUATION ON A TEXT CLASSIFICATION TASK

To perform an analysis analogous to Section 5.2, we turn our attention to a classification problem. We consider the WikiTableQuestions data set introduced by Pasupat & Liang (2015). This data set was derived from tables in Wikipedia articles, and consists of a list of crowdsourced (question, answer, table) triples. The original task is to perform question-answering. We instead construct an artifical task to classify questions into 5 different types.

We analyze a CNN model with essentially the same architecture as the sentiment model from the previous section. The model takes a question (rather than a review) as input, and produces a $5$ label

The result is more depressing than liberating , but it ' s never boring .
On the surface a silly comedy , Scotland , PA would be forgettable if it were n ' t such a clever
adaptation of the bard ' s tragic play .
If there ' s no art here , it ' s still a good yarn -- which is nothing to sneeze at these days .
This method almost never fails him , and it works superbly here .
Well - acted , well - directed and , for all its moodiness , not too pretentious .
This time , the hype is quieter , and while the movie is slightly less successful than the first , it ' s still a
rollicking good time for the most part .
If you want to see a train wreck that you ca n ' t look away from , then look no further , because here it
is .
- West Coast rap wars , this modern mob music drama never fails to fascinate .
The direction has a fluid , no - nonsense authority , and the performances by Harris , Phifer and Cam '
ron seal the deal .
While Scorsese ' s bold images and generally smart casting ensure that `` Gangs '' is never lethargic ,

Figure 6: Sentences with high conductance for filters that have high conductance for the phrase "not bad". These filters are largerly focussed on negation.

| Method | 5 features | 10 features | 15 features | 20 features |
|---|---|---|---|---|
| activations | 54.17 | 84.89 | 91.53 | 92.09 |
| gradient*activation | 73.16 | 90.40 | 91.95 | 91.38 |
| influence | 80.58 | 85.24 | 85.59 | 91.45 |
| conductance | **86.58** | **91.45** | **93.15** | **93.64** |

Table 2: Accuracy of classifiers trained on small number of features selected using the four different methods.

classification (rather than a sentiment score); here each label represents a different type of answer (numeric/date/boolean/string/list).

As in the previous section, we study the feature importances of the filters in the (unique) convolutional layer. We use transfer learning setup similar to the one for object detection (Section 5.2). We used a set of 2831 examples. We split it into two sets – 2265 examples (80%) are used to identify a small number ($\leq 20$)of top filters using one of the four methods (activations, gradient*activations, internal influence and conductance). We train a linear model over these filters. The accuracy of this linear classifier is evaluated on the remaining 566 (20%) examples. The results are presented in table 2. Again, as in the Section 5.2, we see that conductance outperforms the other techniques.

## 7 DISCUSSION

In this paper, we proposed a notion of conductance that extends the notion of attribution to capture the importance of a hidden unit (neuron, or a group of related neurons). We provided a partial axiomatization of neuron importance. We showed empirically that conductance captures the importance of neurons as well as helps interpret their function. We demonstrated the versatility of the method by applying it to a deep network that works with images, and another that operates on text. The method is easy to apply. It does not require the network to be instrumented and can be applied with a few lines of code in your favorite machine-learning framework.

We finish the discussion by describing two ways in which conductance can be used for analyzing deep networks.

**Interpretation** For image networks, hidden-layer neurons (filters) correspond to higher-level features. Identifying important filters could help us provide better explanations of the network's function. For instance, it may be possible to label every filter in the Inception network from Section 5 by hand; for instance Filter $52$ in $mixed3a$ as "glare" (cf. Figure 1 and so on. This could be used to produce a natural language explanation: "Half of the network's prediction was because of shape, and about $20\%$ was because of the glare".

**Rule extraction** A different application of conductance is indicated by Figures 5 and 6. Conductance could help harvest rules for use in a rule-based system. The approach in Section 6.1 seems fairly general: Identify hidden units high conductance for an exemplar of the rule, and then run a large number of examples through the network, extracting phrases that have a high conductance for these high-conductance units.

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

# 8 APPENDIX

## 8.1 PROOF OF PATH METHODS

In this section, we show that given a function $f$ an attribution method that satisfies additivity, insensitivity and completeness is a path method. See section 4.2 for the definition of path method as well as the additivity, insensitivity and completeness axioms.

**Proposition 1.** *For bounded discrete domains, the only attributions methods that satisfy Axioms 1-3 are convex combinations of hidden-layer path methods.*

We follow the proof structure of Wang (1999). First, we show that that any function has a unique representation in terms of a specific basis (Lemma 1). Next we show that the representation satisfies flow constraints (Lemma 2). Finally, we show that the flow can be decomposed into paths (Lemma 4).

### 8.1.1 Definitions

We first define some notation for the discrete case. Let $q \in \mathbb{N}^n$ be a fixed input and $s$ be a fixed baseline. For $a, b \in \mathbb{N}^n$, let $[a, b]$ denote the set $\{t \in \mathbb{N}^n \mid a \leq t \leq b\}$. Let $f : \mathbb{N}^n \to \mathbb{R}$ be a real-valued function on the integer grid, such that $f(s) = 0$. Let $N = \{1, \ldots, n\}$ denote the set of input variables.

Let $1^i$ denote the vector in $\mathbb{N}^n$ whose $i^{th}$ component is 1 and the rest are 0. For a function $f$, we define the *discrete derivative* as $\partial_i f(t) := f(t) - f(t - 1^i)$. For a point $t$, define $A(t) = \{i \mid t_i \neq s_i\}$, the set of active input variables. For a subset $S \subseteq N$ and a vector $t$, let $t_V$ be the restriction of $t$ to $V$. We write $t = (t_V, t_{-V})$ and use $i$ instead of $\{i\}$.

**Remark 3.** *We assume that there is a bounding box $[-Q, Q]$ such that all the cost sharing methods depend only on function values inside the box $[-Q, Q]$ (where $Q$ is some large constant vector $> q$).*

We consider the function $f$ over the box $[-Q, Q]$. We'll assume that $f$ is bounded on the box $[-Q, Q]$.

For a fixed input and baseline, an attribution method $x$ is a mapping from $f$ to $\mathbb{R}^n$, where $x_i(f)$ denotes that attribution to the $i^{th}$ input variable. We consider only the attribution methods that lead to bounded attributions, i.e. $|x_i(f)| < C$ for all bounded functions on $[-Q, Q]$ for some constant $C$.

**Lemma 1.** *(Representation lemma) An attribution method $x$ satisfies additivity, dummy and completeness if and only if, for each $i \in A(q)$, there exists a unique mapping $\mu_i : [-Q + 1^i, Q] \to \mathbb{R}$ such that $|\mu_i(t)| \leq P$ for some constant $P$ and*

$$x_i(f) = \sum_{t \in [-Q+1^i, Q]} \mu_i(t) \partial_i f(t) \text{ for each } f \tag{5}$$

*and*

$$\sum_{i \in A(t)} \sum_{s_{-i} \in [t_{-i}, q_{-i}]} \mu_i(t_i, s_{-i}) = 1 \text{ for each } t \in [-Q, Q] \setminus \{-Q\} \tag{6}$$

*Proof.* The proof of this lemma follows that of Wang's lemma 1, with a couple of key differences. Instead of defining the discrete derivatives $\partial_i(t)$ only for $[s, q]$, we define them over the set $[-Q, Q]$. This allows a path to potentially go outside $[s, q]$, but stay within $[-Q, Q]$. Furthermore, we allow $\mu_i$'s to take negative values to allow the possibility of negative attributions.

To show that $|\mu_i(t)|$ is bounded by a constant for each $t$, we note that $|x_i(f)|$ is bounded. For a fixed $t$, we can construct a function $f$ that is 1 if the $i^{th}$ variable is $\geq t_i$ and 0 otherwise. For this function $x_i(f) = \mu_i(t)$ (other terms in the summation are 0). Since $x_i(f)$ is bounded $|\mu_i(t)|$ must be bounded by a constant. $\square$

Here $\mu_i$ is called the *weight system* associated with attribution method $x$.

**Lemma 2.** *Let $x$ be an attribution method that satisfies additivity and dummy. Let $\mu_i$ be the weight system associated with $x$ (by Lemma 1). Then, for every $t \in [-Q, Q]$ ($t \neq s$ and $t \neq q$), we have:*

$$\sum_{i \in A(q) \cap A(t)} \mu_i(t) = \sum_{i \in A(q) \cap \{i \mid t_i < Q\}} \mu_i(t + 1^i) \tag{7}$$

*and*

$$\sum_{i \in A(q)} \mu_i(s + 1^i) = 1, \tag{8}$$

$$\sum_{i \in A(q)} \mu_i(q) = 1. \tag{9}$$

*Proof.* The proof of this lemma follows closely the proof of Lemma 2 by Sprumont (1999). It uses the above representation lemma for following function:

$$f_t^*(s) = 1 \text{ if } s \geq t \text{ and } 0 \text{ otherwise,}$$

as well as a similar representation lemma for the following function:

$$f_t^{**}(s) = 1 \text{ if } s > t \text{ and } 0 \text{ otherwise}$$

$\square$

Next we describe how to interpret $\mu_i(t)$ at any intermediate point $t$ as flows. Consider the following graph $G = (V, E)$, in which the box $[-Q, Q]$ represents the set of vertices $V$. The set of edges are defined as follows. Let $e_i(t)$ denote the directed edge from $t - 1^i$ to $t$ if $t_i > -Q$ and let $e'_i(t)$ denote the reverse of $e_i(t)$. Let $E(t) = \{e_i(t)\} \cup \{e'_i(t)\}$. and $E = \cup_{t \in [-Q,Q]} E(t)$.

A *f*easible integer flow on $G$ is a set of numbers $\nu(e)$ ($e \in E$) such that:

$$\sum_{e \text{ ends at } t} \nu(e) - \sum_{e \text{ start at } t} \nu(e) = \begin{cases} 1 & \text{for } t = s \text{ (source)} \\ -1 & \text{for } t = q \text{ (sink)} \\ 0 & \text{otherwise} \end{cases} \tag{10}$$

with capacity constraints: $0 \leq \nu(e) \leq C$ for $e \in E$.

**Lemma 3.** *(Garfinkel & Nemhauser (1972)) The constraint matrix defined by the flow constraints in equation 10 is totally unimodular. Furthermore, the extreme points of space of feasible solutions $S = \{y | Ay = b, y \geq 0\}$ are integer vectors.*

**Lemma 4.** *Any attribution method $x$ can be expressed as convex combination of paths $P$.*

*Proof.* First, notice that any path from $s$ to $q$ that stays within the box $[-Q, Q]$ corresponds to a feasible flow – just set $\nu(e) = 1$ for each edge on the path.

To show the converse, we start by showing that the weight system associated with any attribution method satisfies flow constraints. Thus any attribution method corresponds to a feasible flow.

Given a weight system $\mu$, e define a flow as follows. If $\mu_i(t) \geq 0$, then we set $\nu(e_i(t)) = \mu_i(t)$ and $\nu(e'_i(t)) = 0$. On the other hand, if $\mu_i(t) < 0$, we set $\nu(e'_i(t)) = -\mu_i(t)$ and $\nu(e_i(t)) = 0$. In other words, the positive $\mu$'s are treated as flows *i*nto a vertex, while negative $\mu$'s are converted to flows in reverse direction from the vertex.

Now we use Lemma 2. The left hand side of the equation accounts for all the positive flow coming into $t$ plus negative flow leaving $t$, while the right hand side accounts for the positive flow leaving $t$ and negative flow entering $t$. This gives us that the flow constraints in equation 10.

Next, we observe that since the flow on any edge is non-negative and bounded above by $C$, the space of feasible solutions is compact convex set. Hence, any feasible solution can be written as a convex combination of extreme points. In other words, any attribution method is a convex combination of attributions that correspond to the extreme points.

From Lemma 3, we know that all extreme points of the space of feasible solutions are integers vectors. Consider one such integer solution. Such a solution $v$ assigns an integer $v(e)$ to each edge $e$ in the graph. We consider the multiset $M$ of edges, where an edge $e$ is included $v(e)$ times. Furthermore, the flow constraints imply $M$ has an equal number of edges coming into a vertex as leaving it, except for the baseline $s$ which has one extra edge leaving and $q$ which has one extra edge entering. This multiset naturally translates to a path. [5] Thus, we have shown that the weight system $\mu$ is a convex combination of flows that correspond to paths.

This proves that the attribution method $x$ is a convex combination of paths. $\square$

---

[5]This path may contain cycles. Furthermore, multiple paths may correspond to the same integer solution, due to the order in which different cycles are traversed. Notice that the attributions depend only on the integer solution, or equivalently, the multiset $M$. So even if there are multiple possible paths, they all have same attributions.

## 8.2 COUNTER-EXAMPLE FOR BACK-PROPAGATION BASED METHODS

Methods like DeepLift (Shrikumar et al. (2017)), Layer-wise Relevance Propagation (Binder et al. (2016)) and Deep SHAP (Lundberg & Lee (2017)) based on rules for propagating neuron importance backwards from output to input. This process assigns importance to neurons in hidden layers. As a side-effect, this process computes the importance of neurons in the hidden layers. The resulting importance measures for all three methods satisfy Axioms 1-4. However, these methods have an undesirable dependence on the "implementation" of the network. We can vary the implementation of the network in a way that preserves the function computed by a neuron, but changes its importance. This issue was also noticed by Sundararajan et al. (2017) (see section 2.2).

We describe a concrete example below that demonstrates that it is possible to make local changes in a network that leads to a change in importance computed by these methods for other neurons.

Consider a function $f(\cdot) = g(h(\cdot))$, where $h(x_1, x_2) = (x_1, x_2)$ and $g(y_1, y_2) = y_1 \cdot y_2$. We will inspect the neurons in the intermediate layer. Consider the input $(1, 1)$ with the baseline $(0, 0)$ for computing the attributions. Since the function $g(\cdot)$ is symmetric in its inputs, all three methods mentioned above assign equal importance $(1/2)$ to both the hidden neurons.

Now we change the internal layer to replace the second neuron by splitting it as follows: $h'(x_1, x_2) = (x_1, x_2^{1/2}, x_2^{1/2})$ and $g'(y_1, y_2, y_3) = y_1 \cdot y_2 \cdot y_3$. Notice that, the overall function computed is still the same: $g'(h'(x_1, x_2)) = g(h(x_1, x_2)) = f(x_1, x_2)$. But the internal layer has one neuron split into two. Note that $h_1(x_1, x_2) = h'_1(x_1, x_2)$ for all inputs. So, intuitively we expect its importance to stay the same.

However, since $g'(\cdot)$ is a symmetric function of three variables, all of the methods above assign importance value of $1/3$ to each of the three hidden neurons. Thus the importance of first neuron (i.e. $h_1$) changed from $1/2$ to $1/3$ due to a change in the structure of the network on a different path!

