# OpenReview forum: "How Important is a Neuron"
_ICLR.cc/2019/Conference_

### Official Review · AnonReviewer1 · 2018-11-02
**Could use more motivation but it is a good concept.**

**Rating:** 7
**Confidence:** 5

**Review:**

The idea is nice. It is well aligned with tools that are needed to understand neural networks. However, the experiments feel like they are missing motivation as to why this method is being used. The paper does not provide very significant evidence that this method is useful. The negation example is nice but this doesn't seem to display the potential power of the method to understand a neural network.

More motivation for experimental section is needed. If the authors don't discuss a motivation then how will a reader know how to apply the tool? It seems there is no conclusion to take away from the experiments in section 5 (convolutions).

The authors should rethink the structure of the experimental section from the standpoint of convincing someone to use this method. In section 4.1 the authors have a good discussion on what is wrong with other methods in order to motivate their approach but then they don't deliver significant evidence in the later part of the section.

The paper needs more discussion and experiments to explain how and why to use this approach.

While the authors say "attributing a deep network’s prediction to its input is well-studied" they don't compare directly against these methods.

There are many typos and grammar errors

While I think the paper could be much more impactful if the experimental section was greatly reworked; I believe the first 5 pages of the paper are a very good contribution and it should be accepted.

---

> ### Author Response · Authors · 2018-11-21
> **Response to reviewer 1**
>
> We thank the reviewer for their review. The reviewer notes the need to emphasize how and why to use this approach. In the new revision, we have added a discussion section to make a case for this. We will publish the code to compute conductance after the blind-review phase.
>
> The reviewer also mentioned that the paper doesn’t compare directly against various attribution methods. For this, we refer the reviewer to our response for the comment by anonymous.

---

### Official Review · AnonReviewer2 · 2018-11-03
**Requested minor clarifications.**

**Rating:** 7
**Confidence:** 2

**Review:**

The authors propose a notion of conductance to attribute the deep neural network’s prediction to its hidden units. The conductance is the flow of attribution via the hidden unit(s) in consideration. The paper proposes using conductance to not only evaluate importance of hidden unit to the prediction for a specific input but also over a set of inputs. The strongest part of the analysis of conductance is that conductance naturally couples  the path at the base features with that of the hidden layer.

The authors position their work well within the existing approaches in the community and generalizes the efficient use of measuring hidden activation wrt to specific input or set of inputs.

The analysis makes efficient use of mean value theorem in the context of  parametrization of the loss function.

Conductance seems to satisfy the completeness of hidden features. Further, it also satisfies the layer-wise conservation principle with the outputs completely redistributed  to the inputs.

It would be good to see more analysis on the axioms 1 through to 4 for the sake of completeness in the light of partial axiomatization of conductance.

The authors provide empirical evaluation of conductance over a variety of tasks. It would be good to see some more insight in order to relate to interpretability of the importance of neurons, although there has been no claims made on it as its hard to measure importance without interpretability.

---

> ### Author Response · Authors · 2018-11-21
> **Response to reviewer 2**
>
> We thank the reviewer for a detailed review. We agree with the reviewer that a uniqueness result based on the axioms is desirable, but we don’t have it. While we’re able to show that the paths at the input and at the hidden layer must be coupled (i.e. non-oblivious), we just don’t understand the space of non-oblivious methods that well. Mathematically, we don’t have a handle on how the path at the hidden layer can vary as the network below the hidden layer is changed. Partition consistency is the only axiom about the network below the hidden layer, but it is not applicable to all networks. We probably need another axiom to prove uniqueness.
>
> Another key observation made by the reviewer is the interpretability vs importance of neurons. While those are not the same, we demonstrate that conductance can give us some insights about the network (Sections 5.1 and 6.1).

---

### Official Review · AnonReviewer3 · 2018-11-06
**An important measure of Neuron Importance**

**Rating:** 7
**Confidence:** 4

**Review:**

This paper presents a new method to measure the importance of hidden neurons in deep neural networks. The method integrates notions of activation value, input influence to a neuron and neuron influence to the network's output. They provide results confirming that this measure is able to identify neurons that are important for specific tasks.

Quality

The experiments are well designed to verify their hypothesis, although there could be more to make sure those results are not particular to the few selected problems. Nevertheless, the results are consistent across those experiments.

Clarity

The text is well written in general, but the structure could be improved. The introduction contains too much related work, which should be divided in another section. Section 2 contains mostly high level explanations of the work, which should be integrated in the introduction, and thus before the related work section, to improve readability. See minor comments for more specific suggestions.

It is difficult to understand the goal of Section 4.2. Section 2 states that section 4.2 proves that a "path method" must be used in order to satisfy the axioms, but why such axioms are important is not stressed enough. Also, it is not clear why those are called axioms since they are not use to build anything else. It seems to me that those are rather "desirable properties" than axioms.

Originality

A important number of related works are cited and compared with the current work. Although the proposed measure is close to what is proposed by Datta et al., this paper makes the distinction clear and benchmarks its results properly against it.

Significance

There is an increasing need to interpretability of deep neural networks as they get more and more applied to real-world problems. Measures as the one proposed in this paper are a very important building block towards this.

Conclusion

For its original importance measure and the proper experiment benchmarks, I believe this paper should be accepted. There is however many minor issues that should be fixed for the camera-ready version. Although the recommended length is 8 pages, the strict limit is 10, so I would recommended to use a bit of the remaining extra space to conclude the paper properly with a discussion on the results and their consequences, as well as a conclusion to wrap up the paper.

***

Minor Comments

Introduction:
- The term flow is never defined precisely, we need to infer it based on the definition of conductance and attribution.
- First paragraph would be more clear with simple word explanation rather than maths. Also, second sentence is not a complete sentence
- Work on image indicators of importance could be compared better with current work. Indicators can be seen as a measure of importance.
- This sentence is not clear: "[...]; the nature of correlations in the two models may differ".

Section 2:
- Last paragraph of section 2 can be true for any well-performing importance measure. The statements should be put in perspective with others.

Section 3:
- Section 3 should be introduced by explaining the goal of the section otherwise it breaks the flow of reading.
- The role of the baseline x' should be better explained when it is presented (first paragraph of section 3).
- The interchangeable use of the term "conductance of neuron j" for equations 2 and 3 is confusing. Different terms should be use, even if the context makes it possible to infer which one is being referred to.
- Remark 1 seems trivial, but the selection of baseline x' seems less trivial. Some explanations should be devoted to it.
- Second paragraph of remark 1 is not clear. Why couldn't we take another layer's neuron as the neuron of interest, bounding the conductance measure on one layer as the input and the output of the model? If we make the input to be a neuron y rather than the true input x, we could take another neuron z in a subsequent layer to be the neuron of interest, resulting in conductance measure Cond^z_i(y).

Section 4:
- List of importance measure at beginning of Section 4 should probably have citations.
- Backward reference to section 3 seems to be a mistake, should it be subsection 4.2?
- Each of the justifications to get around the issue of distinguishing strange model behavior from bad feature importance technique should be explained briefly in paragraph before section 4.1.

Subsection 4.1:
- I do not understand the problem explained in fourth paragraph of section 4.1. g(f(1 - epsilon)) = 0, why would it be 1- epsilon?
- Problem explained in fifth Paragraph of section is not clear unless what the influence of the unit is clearly stated. Is it simply dg/df?
- A short explanation of what is tested in section 6 should be given at last paragraph of section 4.1. Although the results are favorable to the conductance metric, it is not clear how they precisely confirm the problem of incorrect signs presented in the caricature examples.

Subsection 4.2:
- As said in the my main comments, I am not convinced by the use of the term Axiom. They are not use as building blocks, and are rather used as desirable properties for which the authors prove that only "path methods" can satisfy.
- Footnote 2 on page 5 it difficult to read.
- Although the proof does not seem to use the axioms as a building block, which is fortunate since it would make it a circular argument otherwise, the text suggests so: "Given these three axioms, we can show that:".
- The importance of section 4.2 should be clarified. More emphasis on the importance of the axioms (desirable properties) should be made.

Section 5:
- Choices for experiments should be explained. Why choosing layers mixed** rather than others? Why choosing filters?
- Figures 1-4 are difficult to interpreted on a printed version. Since this is qualitative, I suggest to change the saturation of the images to make them easier to interpret. The absolute values are not important for a qualitative interpretation
- Figure 4 could be more interesting if compared with other classes, like other animal faces. Anyhow, I understand that those were chosen based on the subset of classes used for the experiments.
- Space should be added between figures to better divide the captions

Section 6:
- The difference between experiments of Figure 5 and 6 should be made more clear.

Section 7-8:
- Where are they? No discussion? No conclusion?

---

> ### Author Response · Authors · 2018-11-21
> **Response to reviewer 3**
>
> We thank the reviewer for a detailed review. We have implemented the suggestions for improving clarity.
>
> Regarding our use of ‘axioms’: We follow the economics literature in using axioms as normative concepts, i.e., to denote desirable properties that a neuron importance methods.  And not the use in the mathematical literature, which is to denote statements that are self-evidently true. We have clarified this in the submission.

---

### Public Comment · ~Avanti_Shrikumar1 · 2018-10-30
**Missing citation of "Computationally Efficient Measures of Internal Neuron Importance" by Shrikumar, Su & Kundaje**

"Remark 1" in this paper, which states that a different and more computationally efficient generalization of Integrated Gradients is possible, is directly related to the paper "Computationally Efficient Measures of Internal Neuron Importance" by Shrikumar, Su & Kundaje, published on arXiv on 26th July, 2018 (https://arxiv.org/abs/1807.09946 ). However, the paper by Shrikumar, Su & Kundaje is not cited. The original version of this paper appeared on arXiv on 30th May, 2018 and did not contain Remark 1. It also contained the statement that "computing conductance in tensorflow involved adding several gradient operators and didn't scale very well" in the context of calculating Total Conductance. That statement served as motivation for the work by Shrikumar, Su & Kundaje, and is absent from this version of the work. We therefore request that the authors add a citation to the work by Shrikumar, Su & Kundaje.

---

> ### Author Response · Authors · 2018-10-30
> **Clarification on the missing citation**
>
> First a word of caution for the reviewers: looking up the references mentioned below or in Avanti's remark or what follows will implicitly violate review blindness.
>
> Avanti's comment is partly about a prior version of this work on arxiv. That prior version contained a comment about inefficiency of computing all conductances in a given layer. Our implementation in tensorflow involved adding several gradient operators for this. Any implementation of that requires either doing a sequence of backprop/gradient operations (time inefficiency) or computing Jacobians (space inefficiency). The work by Shrikumar, Su & Kundaje did not address this inefficiency. Instead that work proposed a method similar to Remark 1 in our current submission. We will add a citation to that effect. However, as explained in Remark 1, it lacks an analogue to Equation 2.

---

### Public Comment · (anonymous) · 2018-11-14
**Baselines are cherry-picked**

Echoing a comment made by reviewer 3: in the first line of the paper the authors mention several methods that can be applied to explain a network’s predictions in terms of its inputs. Almost all of these methods can be applied to study neuron-level attribution and many satisfy conservation, including SHAP/DeepSHAP and Layerwise Relevance Propagation. The authors even mention that Layerwise Relevance Propagation satisfies the conservation principle in a footnote on page 5, but they do not compare to it. The baselines are thus very cherry-picked. If the authors wish to make a theoretical argument for their method, that is fine, but selectively leaving out some of the strongest methods in the benchmarking is very misleading because it gives the impression that the methods in the benchmark are the only methods that could be used for this purpose. SHAP/DeepSHAP in particular are quite widely used and should be compared to.

---

> ### Comment · AnonReviewer3 · 2018-11-15
> **Concerns about related works and baselines.**
>
> From what I understand, SHAP/DeepSHAP and Layerwise Relevance Propagation gives some form of importance measure of the input features. The experiments described in sections 5 and 6 are designed to study how importance measures of the hidden neurons captures informations such as class selectivity or negation selectivity. I believe this is why the authors did not compare with SHAP/DeepSHAP and related methods. They could be compared however by considering the hidden neuron activations as the input of the network. This would also be an important baseline, further showing that ignoring information prior to the selected hidden neurons is detrimental for the importance measure, assuming that their method would still have the best results.
>
> My main concerns with the experiments however is more about the limitation of the setups than about the limitation of baselines as I explained in my review. Results on only two different problems with a single architecture for each is not a strong proof of the generality of the measure. Each problem and architecture are very different which could help showing the generality, but the analyses are different in nature, one is quantitative while the other one is qualitative, which makes them hardly comparable.

---

> > ### Author Response · Authors · 2018-11-21
> > **Clarification on baselines**
> >
> > We thank the anonymous commentator and reviewer #3 for the critiques about benchmarks.
> > We respond to the two types of issues with our comparisons:
> >
> > (a) Comparing against other methods: We chose our benchmarks to be published techniques of neuron importance. In this sense, we certainly make no effort to cherry pick. However, we do agree with the observation that methods such as LRP, DeepLift, DeepShap could be used as measures of neuron importance, though they have not been proposed for this purpose.
> >
> > To partially address this, we performed a new theoretical analysis of DeepShap/LRP (see Appendix 8.2 and Section  4.1.) . We find that the importance measures from these methods have an intuitively undesirable dependence on the “implementation” of the network. That is,  you can vary the network architecture in a way that neuron computes the very same function, but the feature importance changes.
> >
> > (b) Evaluating on different tasks: First, we should point out that we do have a quantitative analysis on *both the text task and the image one. (see sections 5.2 and 6.2).  We also have qualitative insights on both tasks (Sections 5.1 and 6.1). It is also true that sections 6.1 and 6.2 are not quite the same task. But this was mostly to make a quantitative eval of feature selection possible. Finally, note that we picked two large-scale, practically applied networks. Not MNIST and not some toy task.
> >
> > That said, we do agree with Reviewer #3 that the empirical evaluations are not strong proof of generality. Hopefully our theoretical arguments compensate for this. As we all know, there is no natural ground truth for judging attribution or neuron importance. So almost every empirical evaluation of attribution has some issue. But all in all, we do agree there is room for more empirical evaluation.

---

> > > ### Comment · AnonReviewer3 · 2018-11-26
> > > **Confirming point (b)**
> > >
> > > I confirm that section 5.2 and 6.2 both have quantitative analyses. I missed the table 2 while responding to the anonymous comment.

---

### Meta-Review · Area_Chair1 · 2018-12-15
**Interesting work on how to measure the importance of an individual neuron in a network**

**Confidence:** 5
**Recommendation:** Accept (Poster)

**Metareview:**

This paper proposes a new measure to quantify the contribution of an individual neuron within a deep neural network. Interpretability and better understanding of the inner workings of neural networks are important questions, and all reviewers agree that this work is contributing an interesting approach and results.